# Neutrophil Extracellular Traps Drive Dacryolithiasis

**DOI:** 10.3390/cells12141857

**Published:** 2023-07-14

**Authors:** Leticija Zlatar, Thomas Timm, Günter Lochnit, Rostyslav Bilyy, Tobias Bäuerle, Marco Munoz-Becerra, Georg Schett, Jasmin Knopf, Jens Heichel, Mohammad Javed Ali, Mirco Schapher, Friedrich Paulsen, Martin Herrmann

**Affiliations:** 1Department of Internal Medicine 3—Rheumatology and Immunology, Universitätsklinikum Erlangen, Friedrich Alexander University Erlangen-Nürnberg (FAU), 91054 Erlangen, Germany; marco.munozbecerra@uk-erlangen.de (M.M.-B.); georg.schett@uk-erlangen.de (G.S.); jasmin.knopf@medma.uni-heidelberg.de (J.K.); martin.herrmann@uk-erlangen.de (M.H.); 2Deutsches Zentrum für Immuntherapie (DZI), Universitätsklinikum Erlangen, Friedrich Alexander University Erlangen-Nürnberg (FAU), 91054 Erlangen, Germany; 3Institute of Biochemistry, Justus-Liebig University Giessen, 35392 Giessen, Germany; thomas.timm@biochemie.med.uni-giessen.de (T.T.); guenter.lochnit@biochemie.med.uni-giessen.de (G.L.); 4Department of Histology, Cytology, Embryology, Danylo Halytsky Lviv National Medical University, 79010 Lviv, Ukraine; r.bilyy@gmail.com; 5Institute of Radiology, Preclinical Imaging Platform Erlangen (PIPE), Universitätsklinikum Erlangen, Friedrich Alexander University Erlangen-Nürnberg (FAU), 91054 Erlangen, Germany; tobias.baeuerle@uk-erlangen.de; 6Department of Pediatric Surgery, University Medical Center Mannheim, University of Heidelberg, 68167 Mannheim, Germany; 7Department and Policlinic of Ophthalmology, Martin Luther University of Halle-Wittenberg, 06108 Halle, Germany; jens.heichel@uk-halle.de; 8Govindram Seksaria Institute of Dacryology, L.V. Prasad Eye Institute, Road No 2, Banjara Hills, Hyderabad 500034, India; javed@lvpei.org; 9Institute of Functional and Clinical Anatomy, Friedrich Alexander University Erlangen-Nürnberg (FAU), 91054 Erlangen, Germany; friedrich.paulsen@fau.de; 10Department of Otorhinolaryngology, Head and Neck Surgery, Universitätsklinikum Erlangen, Friedrich Alexander University Erlangen-Nürnberg (FAU), 91054 Erlangen, Germany; mirco.schapher@klinikum-nuernberg.de; 11Department of Otorhinolaryngology, Head and Neck Surgery, Paracelsus University, 90419 Nürnberg, Germany

**Keywords:** mucopeptide concretions, dacryoliths, dacryolithiasis, lacrimal sac, neutrophils, neutrophil extracellular traps

## Abstract

Mucopeptide concretions, previously called dacryoliths, are macroscopic stones that commonly obstruct the lacrimal sac. The mechanism behind dacryolithiasis remains unclear; however, the involvement of various immune cells, including neutrophils, has been confirmed. These findings remain limited, and no information on neutrophil extracellular traps (NETs), essentially involved in the pathogenesis of other lithiases, is available yet. Here, we employ microcomputed tomography, magnetic resonance tomography, histochemistry, mass spectrometry, and enzyme activity analyses to investigate the role of neutrophils and NETs in dacryolithiasis. We classify mucopeptide concretions into three types, with respect to the quantity of cellular and acellular material, polysaccharides, and mucosubstances. We propose the role of neutrophils and NETs within the existing model of gradual formation and growth of mucopeptide concretions, with neutrophils contributing to the initial stages of dacryolithiasis, as they localized on the inner (older) parts of the tissue. As NETs localized on the outer (newer) parts of the tissue, we link their role to the late stages of dacryolithiasis, presumably maintaining the proinflammatory environment and preventing efficient clearance. An abundance of IgG on the surface indicates the involvement of the adaptive immune system later as well. These findings bring new perspectives on dacryolithiasis, in which the innate and adaptive immune system are essentially involved.

## 1. Introduction

Mucopeptide concretions or dacryoliths (MPC-D), formerly also referred as “mucoliths”, are macroscopic stones that are formed in the lacrimal sac and the nasolacrimal ducts [1,2]. Unlike other ‘liths’ in several parts of the human body, MPC-Ds are mostly composed of organic material, and their pathophysiology remains unclear [3]. However, some predisposing factors that contribute to their formation have so far been described. These include smoking, primary acquired nasolacrimal duct obstruction (PANDO) [3,4], and previous chronic dacryocystitis [3]. The effects of age and gender remain elusive [2], and some even claim that the age, duration of epiphora, history of acute dacryocystitis, or previous use of medications have no effect on dacryolithiasis [4]. An alteration in the production of mucins and trefoil factors TFF1 and TFF3, as well as the induction of TFF2, have been established as major risk factors for dacryolithiasis [1,2]. Under physiological conditions, trefoil factors protect the mucous epithelia post injury by supporting the trapping of viruses with the help of immunoglobulins, fostering cell migration to the place of injury, or displaying generally antiapoptotic effects [5,6,7]. Under pathological conditions, such as during dacryolithiasis, the production of trefoil factors and cosecretion of mucins is significantly increased. Two major types of lacrimal system concretions exist; mucopeptide and bacterial. They differ in their location and histopathologic composition. A third category of intermediate concretions, with characteristics of both of these types, has also been described [8]. The formation of bacterial concretions has been classically attributed to *Actinomyces* species. Recently, *Streptococcus* and *Staphylococcus* species have also been reported as major genera involved in their formation. Other bacteria, and also fungi, were previously isolated from bacterial concretions [9]. Not surprisingly, epithelial cells, neutrophils, T and B lymphocytes, as well as macrophages, have been found in MPC-Ds as well. [1,10]. Ali and Paulsen performed several electron microscopic, immunohistochemistry, and cinematic rendering techniques to determine that the initial pathogenic event is a microscopic trauma within the lacrimal drainage and blood clotting in the nidus which initiates the dacryolithiasis in a predisposed individual [11,12,13]. Subsequently, the local mucopeptides and those in the tears sequentially lay mucopeptides around this nidus.

Neutrophils are immune cells involved in phagocytosis, degranulation, reactive oxygen species (ROS) and cytokine production, and in neutrophil extracellular trap (NET) formation. They can be found in blood in high quantities, and migrate to the site of infection to initiate the immune response [14]. A well-known mechanism of bacterial killing by lysozyme highlights the importance of neutrophil antimicrobial proteins in host defense against pathogens [15]. Within cytoplasmic granules, neutrophils contain various serine proteases, including neutrophil elastase (NE), cathepsin G, and proteinase 3 (PR3) [16,17]. Neutrophil granules also contain azurocidin, an antimicrobial protein similar in structure to the three aforementioned serine proteases, with important roles in the host defense and inflammation [18]. Within secondary granules, neutrophils contain lactoferrin, the first line of defense against microbial infections [19]. S100A8 and S100A9 proteins are constitutively expressed in neutrophils in high levels, and comprise up to 45% of all cytoplasmic proteins. They act as calcium binding proteins under physiological conditions; however, during inflammation, their upregulation modulates cytokine secretion [20,21]. Another proinflammatory molecule expressed by neutrophils is resistin; it also has antimicrobial function [22]. In addition, defensins, cationic antimicrobial peptides, are produced by neutrophils [23].

Neutrophils are well-known for forming NETs, whose primary function is to trap and kill various pathogens. To detect NETs, antibodies against granular proteins, such as NE and myeloperoxidase (MPO), as well as citrullinated, decondensed chromatin, using citrullinated histone H3 (citH3) are commonly used in combination with various DNA intercalating dyes, such as DAPI, Sytox Green, Hoechst 33342, and PI [24,25]. In the context of the immune system, NETs are “double edged swords”, meaning they can act pro- or anti-inflammatories [26], depending on the density of neutrophil infiltration [27]. Eventually, serum DNases degrade NETs, and phagocytes engulf them. However, an imbalance in NET formation and clearance leads to pathogenesis of various diseases, and this balance needs to be strictly maintained [28]. Importantly, the involvement of NETs in the pathology of other stone diseases has already been confirmed. Examples include cholelithiasis or sialadenitis, in which gallstone formation or salivary gland stone formation, respectively, occur. In such pathologies, NETs act as the “initiators” for the development and growth of sialoliths or gallstones, respectively [29,30].

In this study, we investigate the link between neutrophils and NETs and human MPC-Ds. Even though heterogenous in shape and size, all samples studied here obtained the shape of the lacrimal sac in which they were formed. They displayed a stratified structure and contained only small amounts of electron dense inclusions. We classified them into three different types based on the amount of cellular and acellular material, polysaccharides, and mucosubstances. Using immune fluorescence (IF) for various neutrophil and NET associated markers (citH3, MPO, and NE), we examined the role of neutrophils and NETs in dacryolithiasis in the context of their localization within the tissue. We found neutrophil- and NET-associated markers at different quantities in all samples, and also detected high NE activity in those samples with a high expression of NE. Importantly, NETs localized on the surface of the samples, linking NET formation to the later stages of disease pathogenesis. We propose that NETs are rather involved in later stages and might contribute to their aberrant clearance via a yet-undescribed mechanism. We performed stainings of various other components of the innate and adaptive immune system such as hemoglobin or IgG, respectively. In general, we found all antigens at various expression levels, however, at higher quantities on the surface of the samples. Taken together, data presented here indicate that processes related to neutrophils, NETs, and the adaptive immune response occur during the formation, growth, or maintenance of MPC-Ds.

## 2. Materials and Methods

### 2.1. Ethical Statement

Investigations on human material were performed in accordance with the Declaration of Helsinki and with the approval by the ethical committee of the University Hospital Erlangen (permit number 243_15 B). Informed written consent about the use of tissue samples was given by each patient.

### 2.2. Human Tissue Samples

Seven mucopeptide concretions (MPC-Ds) were removed via dacryocystorhinostomy (DCR) and processed for histological analyses. Paraffin-embedded tissue sections (6 µm) were prepared at the Institute of Functional and Clinical Anatomy, Erlangen, Germany and sent to the Department of Medicine 3—Institute for Rheumatology and Immunology, Erlangen, Germany, for further evaluation.

### 2.3. Magnetic Resonance Tomography (MRT)

MRT Imaging of MPC-Ds was performed at the Institute of Radiology at the Preclinical Imaging Platform (PIPE), Erlangen, Germany.

### 2.4. Microcomputed Tomography (µCT)

µCT analysis of the MPC-Ds was performed at the University Hospital Erlangen, Department of Medicine 3—Institute for Rheumatology and Immunology, Erlangen, Germany as described elsewhere [30].

### 2.5. Mass Spectrometry (LC-ESI-MS)

We deparaffinized 10 µm thick sections of MPC-Ds. Using a scalpel and a light microscope, we scraped the inner part (core) of the tissue from the glass slide into a sample tube. Two sections per sample were prepared in this way, pooled together, and sent for complete proteome analysis (MALDI) in dry form to the Institute of Biochemistry, Justus-Liebig University Giessen, Giessen, Germany as described below:

Tryptic digestion of proteins—samples were dissolved in lysis buffer (6 M urea (Sigma, Taufkirchen, Germany), 2 M thiourea (Sigma), 4% 3-3′-(Cholamidopropyl)-3,3-dimethylammoniumpropylsulfat (CHAPS; Roth, Karlsruhe, Germany), 30 mM dithiothreitol (DTT; Fluka, Seelze, Germany), 2% IPG-buffer pH 3–10 (GE Healthcare, Freiburg, Germany), and digested following the FASP protocol [31]. Tryptic peptides were acidified using 1% TFA and purified using a C18-ZipTip (Millipore, Burlington, MA, USA), dried under vacuum, and finally dissolved in 10 µL of 0.1% TFA.

Liquid-Chromatography Electrospray-Ionization Mass Spectrometry (LC-ESI-MS)—for analysis, 1 µg of the sample was loaded onto a 50 cm µPACTM C18 column (Pharma Fluidics, Gent, Belgium) in 0.1% formic acid (Fluka) at 35 °C. Peptides were eluted with a linear gradient of acetonitrile from 3% to 44% over 240 min followed by a wash with 72% acetonitrile at a constant flow rate of 300 nL/min (ThermoScientific™UltiMate™3000RSLCnano) and infused via an Advion TriVersa NanoMate (Advion BioSciences, Inc., New York, NY, USA) into an Orbitrap Eclipse Tribrid mass spectrometer (ThermoScientific). The mass spectrometer was operating in positive-ionization mode with a spray voltage of the NanoMate system set to 1.5 kV and source temperature at 275 °C. Using the data-dependent acquisition mode, the instrument performed full MS scans every 3 s over a mass range of *m/z* 375–1500, with the resolution of the Orbitrap set to 120,000. The RF lens was set to 30%; auto gain control (AGC) was set to standard with a maximum injection time of 50 ms. In each cycle the most intense ions (charge state 2–7) above a threshold ion count of 50,000 were selected with an isolation window of 1.6 *m/z* for HCD-fragmentation at normalized collision energy of 30%. Fragment ion spectra were acquired in the linear IT with a scan rate set to rapid and mass range to normal and a maximum injection time of 100 ms. After fragmentation, the selected precursor ions were excluded for 15 s for further fragmentation.

Data acquisition and analysis—data were acquired using Xcalibur 4.3.73.11. (Thermo Fisher Scientific, Waltham, MA, USA) and analyzed using Proteome Discoverer 2.5.0.400 (Thermo Fisher Scientific). The Mascot search engine 2.8.2 (Matrix Science) was used to search against the Swissprot_humandatabase (v. 2022_04, sequences = 568,363, residues = 205,318,884, homo sapiens sequences = 20,402). A precursor ion mass tolerance of 10 ppm was used, and one missed cleavage was allowed. Carbamidomethylation on cysteines was defined as a static modification with optional oxidation of methionine. The fragment ion mass tolerance was set to 0.8 Da for the linear IT MS2 detection. The FDR for peptide identification was limited to 0.01 by using a decoy database.

### 2.6. Neutrophil Elastase (NE) Activity Measurement

We deparaffinized sections of MPC-Ds, scratched each tissue from the glass slide into an Eppendorf tube, and added 600 µL Dulbecco’s Phosphate Buffered Saline (DPBS, Thermo Fisher Scientific). We thoroughly resuspended all samples, and pipetted 180 µL triplicates into a 96-well plate. A total of 5 UN/mL elastase from human leukocytes (Sigma-Aldrich, Burlington, MA, USA, E8140) was used as a positive control. We added 20 µL (100 µM) of the fluorogenic substrate MeOSuc-AAPV-AMC (Santa Cruz Biotechnology, Dallas, TX, USA, sc-201163) to all wells, and measured the neutrophil elastase activity at 37 °C for 12 h at a 10 min interval by employing the TECAN Infinite 200 Pro fluorescence plate reader (Tecan, Männedorf, Switzerland) with a filter set of excitation at 360 nm and emission at 465 nm.

### 2.7. Histochemistry/Immune Fluorescence (IF)

We processed the histological sections by melting and washing off paraffin, then performed hematoxylin and eosin (HE) or Periodic acid-Schiff (PAS) staining, or fluorescence stainings employing the following primary antibodies (all rabbit, antihuman): neutrophil elastase (NE, Abcam, Cambridge, UK, ab68672, 1:100), myeloperoxidase (MPO, Abcam, ab9535, 1:200), citrullinated histone H3 (citH3, Abcam, ab5103, 1:300), protein arginine deiminase 4 (PAD4, Sigma, P4749, 1:50), Aquaporin 9 (AQP-9, Abcam, ab84828, 1:100), hemoglobin (α subunit, Abcam, ab92492, 1:100), immunoglobulin G (IgG, Southern Biotech, Birmingham, AL, USA, 6140-01, 1:100), Fetuin A (ThermoFisher Scientific, PA5-51594, 1:100), and fibrinogen (Agilent Technologies, Santa Clara, CA, USA, A0080, 1:500). We used IgG goat antirabbit Cy5 (Jackson Lab., Bar Harbor, ME, USA, 111-175-144, 1:400) as a secondary antibody. We further stained mucins with primary antibodies against Mucin 5AC (MUC5AC, Sigma-Aldrich, MAB2011, 1:1000) or Mucin 5B (MUC5B, Sigma-Aldrich, MABT899, 1:50), both mouse antihuman. As a secondary antibody, we used IgG goat antimouse Cy5 (Jackson Lab., 115-175-146, 1:400). We stained various cytokeratins with rabbit antihuman polyclonal antibodies: Cytokeratin 1 (CK-1, ThermoFisher Scientific, PA5-119070, 1:100), Cytokeratin 2 (CK-2, Bio-Techne, Minneapolis, MN, USA, NBP1-31423, 1:100), Cytokeratin 9 (CK-9, ThermoFisher Scientific, PA5-24783, 1:100), and Cytokeratin 10 (CK-10, Bio-Techne, NBP1-85604, 1:1000). We employed IgG goat antirabbit Cy5 (Jackson Lab., 111-175-144, 1:400) as a secondary antibody. In addition, we performed immunostaining of glycosylated proteins Galactose-β-2,6-Sialic acid (SNA) or Galactose-β-1,3-GalNAc (GalNAc), using directly labeled lectins: *Sambucus nigra* lectin from Elderberry Bark conjugated with Cy5 (Vector Laboratories, Newark, CA, USA, Cl-1305, 1:50) or *Ricinus communis* agglutinin I conjugated with Rhodamine (Vector Laboratories, RL-1082, 1:250).

As a counter-stain, various DNA dyes were used: Hoechst 33,342 (Molecular Probes, 0.2 µg/mL), 4′,6-Diamidin-2-phenylindol (DAPI, 0.2 µg/mL), propidium iodide (PI, Sigma-Aldrich, 2 µg/mL), or Sytox Green (SG, Thermofisher, 1:10 000). DNA was additionally stained with primary IgM antibody raised in mice (a-DNA, Merck Millipore, Darmstadt, Germany, CBL186, 1:100) and secondary IgM antibody goat antimouse TRITC (Jackson Lab., 115-025-020, 1:400). Controls were stained with DNA stain and a corresponding secondary antibody only. All stainings were embedded in DAKO fluorescence mounting medium (Agilent, Santa Clara, CA, USA, S3023).

Thioflavin T (ThT) staining was performed as follows: 3 mM ThT dissolved in 30:70 Ethanol; DPBS (ThermoFisher Scientific) was added to the deparaffinized tissue sections for 20 min at room temperature. The sections were then washed in 70% Ethanol, followed by washing in DPBS (ThermoFisher Scientific), and finally mounted using the DAKO fluorescence mounting medium (Agilent, S3023) and dried overnight.

Images obtained from PAS staining or various DNA stainings were analyzed using morphometry in Photoshop CC 2018 (Adobe, Munich, Germany). The mean fluorescence intensity (MFI) was extracted and exported in an R data table object. The FIt-SNE dimensionality reduction was performed using the function ‘run.fitsne()’ from the Spectre R package (10.1002/cyto.a.24350), with instructions and the source code provided at https://github.com/ImmuneDynamics/spectre, accessed on 11 November 2022. Finally, we performed the image conversion to Rainbow RGB in Fiji.

### 2.8. Macroscopy and Fluorescence Microscopy

We took macroscopic images using a Nikon 700 camera (Nikon, Tokyo, Japan) with a CMOS sensor in FX format (36.0 × 23.9 mm and 12.87 million pixels) and performed fluorescence microscopy employing the fluorescence scanner (Aperio Versa 8, Leica Biosystems). We processed the obtained images in Photoshop CC 2018 (Adobe, Munich, Germany).

## 3. Results

We first analyzed the morphological features of human MPC-Ds via macrophotography (Figure 1a), magnetic resonance tomography (MRT, Figure 1b), and microcomputed tomography (µCT, Figure 1c). We observed differences between MPC-Ds in both shape and size, but they commonly confirmed to the shape of the lacrimal sac in which they were formed. They were stiff and their color varied from yellow to orange (Figure 1a). MRT imaging and 3D surface reconstruction revealed a stratified structure (Figure 1b). Employing the µCT analysis, we observed only a few electron dense inclusions spread across the whole body of the MPC-D (Figure 1c). This indicates limited calcifications.

Seven MPC-Ds obtained from both female (29%) and male (71%) patients with an average age of 63 (46–85) years, were analyzed via histochemistry. We employed hematoxylin and eosin (HE) staining to differentiate between cellular and acellular regions (Figure 2a). HE staining revealed morphological differences between the seven MPC-Ds tested. Accordingly, we divided them into three groups: type I, characterized by intense hematoxylin staining of chromatin/cells’ nuclei (blue to purple); type II, characterized by predominant eosin staining of acellular material (pink); and type III, characterized by an equal staining of hematoxylin and eosin, derived from a mixture of chromatin and non-nuclear material (purple to pink). Nonetheless, we observed regions containing cells in all samples. Macrophotographs of paraffin-embedded tissue sections are displayed in Figure 2b. We further employed Periodic acid-Schiff (PAS) staining to detect polysaccharides and mucosubstances. Figure 2c displays a FIt-SNE plot based on the PAS staining. Different clusters can be observed: (I) P93, (II) P94 and P102, (III) P95 and P99, (IV) P96 and P101, (V) P97, and (VI) P98 and P100.

We next performed complete proteomic analysis of human MPC-Ds, employing the LC-ESI-MS technique (Figure 3a). To examine the composition of MPC-Ds in the early stages of their formation, only the inner (core) parts were analyzed. The analysis revealed an abundance of neutrophil-associated markers in all samples tested: S100A9, cathepsin G (CTSG), lactoferrin (LTF), lysozyme (LYZ), resistin (RETN), neutrophil defensin 1 (DEFA1), and myeloperoxidase (MPO). The samples also contained average to high amounts of S100A8, neutrophil elastase (ELANE), and azurocidin (AZU1). Some proteins were found in either low quantities, or not at all: eosinophil cationic protein (RNASE3), lipocalin-1 (LCN1), zymogen granule protein 16 homolog B (ZG16B), defensin alpha 4 (DEFA4), neutrophil-gelatinase-associated lipocalin (NGAL), S100A12, cathelicidin antimicrobial peptide (CAMP), grancalcin (GCA), bactericidal permeability-increasing protein (BPI), S100P, and matrix-metalloproteinase 8 (MMP8).

As we found many proteins characteristic to neutrophils, we further analyzed the obtained tissues sections for NE activity (Figure 3b). NE activity was detected in all MPC-Ds tested, at different levels. We observed the highest NE activity for samples P102 and P96, robust activity for samples P99 and P95, and lower, but nevertheless considerable, NE activity for samples P94, P97, and P98. Of note, NE in the positive control (5 mU NE enzyme) had different kinetics than the NE in the MPC-Ds.

We then performed fluorescence stainings of various proteins, some of which were also found via proteomic analysis, to determine their localization within the tissue. A part of the tissue from one representative stone (P99) is shown in Figure 3c. Fluorescence stainings revealed a high signal for PAD4, IgG, GalNAc, SNA, hemoglobin, and CK-1. Table 1 contains mean values for each antigen in all MPC-Ds, obtained via the visual estimation of the fluorescence stainings. Some antigens were found in less than half of the samples (NE, citH3), and the rest were found in the majority (MUC5B, Fetuin A, MPO, MUC5AC, CK-1, fibrinogen, AQP-9, CK-9) or all of the samples (hemoglobin, CK-2, SNA, IgG, PAD4 and GalNAc). All antigens were predominantly found on the surface of the MPC-Ds, except for hemoglobin which was found in higher quantities in the inner (older) parts of MPC-Ds. The expression levels of different antigens varied considerably among the seven MPC-Ds tested (Appendix A). Interestingly, in those samples in which little to no neutrophil markers were detected by IF, but found by LC-ESI-MS (P94, P96, P97, and P98), we detected a positive ThT signal (Figure 3d). This indicated an abundance of amyloid fibrils, i.e., filamentous protein aggregates that sterically hinder the accessibility of the antigens for detection by antibodies.

Lastly, various DNA dyes, including propidium iodide (PI), Hoechst33342, and DAPI, as well as a DNA antibody (anti-DNA), were used to detect DNA in MPC-Ds (Figure 4). Here, different tissue fragments, displayed as single dots, similar to each other in terms of DNA staining, clustered (group) together. The anti-DNA staining differed from all other DNA stainings, and preferentially stained the scattered DNA (Figure 4b), marked as region I in the schematic representation of the sample P94 (Figure 4a); such DNA can be found in NETs. FIt-SNE plots obtained by plotting the fluorescence intensity of separate DNA stainings are depicted in Figure 4d (sample P94) and Figure 4f (fluorescence intensity of all DNA stainings for all samples). No differences in the density of the DNA fluorescence was observed (Figure 4c). The DNA staining of MPC-Ds from different patients was generally heterogenous (Figure 4e).

Since we discovered an abundance of neutrophil-associated markers, we examined the tissue for neutrophils and NETs via fluorescence microscopy. A representative sample, abundant in both NE and MPO in the MALDI analysis and detected by IF, is shown in Figure 4g. We also observed an abundance of NE in the following MPC-Ds: P95, P99, and P102 as confirmed by robust NE activity (Figure 3b), high NE fluorescence signal (Appendix A), and high MASCOT scores obtained in proteomic analysis (Figure 3a). The colocalization of NE and DNA appeared exclusively on the surface of the sample, indicating NET formation. We detected NE in the core of the sample, however, without DNA colocalization. The staining in which antibody against DNA was used (anti-DNA), shows a different staining than DAPI, confirming that extracellular DNA from NETs can only be found on the surface of the sample.

## 4. Discussion

Mucopeptide concretions (MPC-Ds) are known to take the shape of the lacrimal sac and nasolacrimal duct, in which they are formed [12]. Findings presented here are in line with previous publications, which reported the heterogeneity of MPC-Ds [2]. All samples we analyzed differed in size and shape. They obtained the form of the lacrimal sac and contained very few electron dense inclusions, indicating low number of calcifications. This confirmed that the MPC-Ds contain low amounts of inorganic material in the form of minor calcifications and are mostly composed of organic material and biological components [3]. Their layered, onion-like structure points towards gradual formation and growth [13].

In 2006, Paulsen et al. performed an extensive study on the composition of dacryoliths. None of the stones examined revealed any calcification under X-ray examination. HE staining, as well as PAS staining, were then performed, revealing a structure made of lobes and lobules, built on amorphous core material with scant cellular material. The amorphous material had either the appearance of debris or was organized in different layers [1]. Here, we define this as only one type of the MPC-Ds, considering there are other types, semirich or rich in cellular material, as well. Therefore, type I, II, and III MPC-Ds are described, based on sample morphology. The amount of cellular infiltration could indicate different extents of inflammation within the lacrimal sac obstructed by MPC-Ds. Even though stainings for neutrophils were performed at that time, only neutrophil antimicrobial substances (defensins) were investigated. Neutrophils were present in impressive amounts in most, but not all, of the investigated dacryoliths [1]. In line with these findings, we found defensins, secretory products of neutrophils, as well. In addition, we employed stainings for NE, citH3, and MPO together with DNA to differentiate between neutrophils and NETs within MPC-Ds. A recent publication by Wang et al. reports the superiority of the anti-DNA-IgM antibody in detecting loose, decondensed DNA, such as that in the patches of NETs [25]; we used this novel method to detect extracellular DNA in dacryoliths. We found neutrophils on the inside of the tissue and NETs on the rim of the tissue, i.e., NE colocalizing with the DNA. This indicated that, whereas neutrophils are involved in the early stages of the formation and growth of MPC-Ds, NETs play a role in disease pathogenesis later on. By localizing on the surface of the MPC-Ds, they might prevent their clearance and maintain the proinflammatory environment.

In 2018, Ali et al. showed via electron microscopy that the inside (core) of the MPC-Ds consist of extensive fibrillary network, rich in red blood cells. The presence of granulocytes and epithelial cells has been described as occasional here [12]. Accordingly, when we analyzed MPC-Ds, we investigated the core separately from the surface. Using IF, we detected hemoglobin in the core of the dacryolith as well, the only antigen studied found in higher quantities in the core of the sample, rather than the surface. This also supports the current hypothesis, where blood leaking is considered as the first step in dacryolithiasis, preceded only by a trauma of mechanical or chemical type [12]. Interestingly, we also observed hemoglobin on the surface of the samples, pointing toward possible blood leaking in later stages of dacryolithiasis as well. In line with these findings, we observed the occasional presence of granulocytes in the core of the concretions using IF, but an abundance of granulocyte-derived peptides and protein fragments using mass spectrometry.

We initially hypothesized that the reason for this was the inability of the antibody to physically reach to the antigen due to a dense/compact and, therefore, impenetrable network of cytokeratins. Therefore, IF stainings for four different cytokeratins (CK-1, CK-2, CK-9, and CK-10) were performed. We detected cytokeratins in almost all samples, which confirmed the presence of epithelial cells within MPC-Ds [32]. We mainly found cytokeratin 2 (CK-2) and cytokeratin 9 (CK-9). CK-1 localized mostly on the rim of the tissue; CK-2 and CK-9 were found in other (inner) parts of the tissue. CK-10 was the least abundant cytokeratin. Mucins and TFF peptides have been described as major components of MPC-Ds; however, reactivity against MUC5AC and MUC5B was observed in most cases, but not all [1]. No reactivity for mucins in some cases was attributed to a lack of cells in the section. Accordingly, we did not detect mucins in high quantities and in all samples tested; their distribution was rather inhomogeneous. We, therefore, stained the tissue for glycosylated proteins (GalNAc and SNA), as mucins are often glycosylated and TFF activity depends upon their glycosylation state [33]. The discrepancies between low mucins signal and high glycosylation signal indicated that mucins might be present but could not be detected by immune fluorescence. Considering that all MCP-Ds in our case contained at least some cellular regions, we contribute this to the aggregation of proteins into amyloid-like structures, which form a dense network and cause the impermeability of the tissue for antibody binding. We confirmed this by the use of Thioflavin T (ThT), a cationic benzothiazole dye which increases in fluorescence upon binding to amyloid fibrils (filamentous protein aggregates) [34,35], commonly used in histology and for protein characterization. With ThT staining, one can detect the conformationally altered, folded intermediates, and aggregated or fibrillated proteins, which can still be immunogenic [36]. We detected a strong ThT signal only for those samples in which the antigens could not be detected by IF (P94, P96, P97, and P98). In addition, the correlation between the HE staining and NE activity is not a direct one. A reason for this is that not all infiltrating cells seen in HE staining correspond to neutrophils, rather other immune cells as well. Considering, however, the immense heterogeneity of the samples analyzed, our classification based on HE and PAS stainings appeared most appropriate. Further studies with larger numbers of patient samples might bring some clarity to the pattern of heterogeneity, such as age or gender.

The first proteomic analysis of MPC-Ds (dacryoliths) in the context of rebamipide treatment was conducted in 2018 as well, identifying the proteins present in MPC-Ds [37]. Here, we link these findings to the localization of various antigens in the tissue, employing IF. By performing proteomic analysis, we observed an abundance of MPO in all MPC-Ds, with the highest expression in tissues P96 and P102. In line with this, we also found MPO in all samples when we performed IF stainings. Accordingly, MPO expression was highest in sample P102. However, we observed a discrepancy between the two methods for sample P96. MPO was found in high quantities in proteomic studies; however, it was not present in IF staining at all. Employing MALDI, we detected NE in all MPC-Ds, with the lowest expression in P94 and P97, in line with the IF stainings. Two other samples, P96 and P98, were poor in NE in IF stainings. We contribute these discrepancies to the aforementioned dense network of amyloid fibers, confirmed by the ThT staining. In line with previous findings, we found abundant S100A9, S100A8, lysozyme C, neutrophil defensin 1, cathepsin G, azurocidin, neutrophil elastase, eosinophil cationic protein, myeloperoxidase, and lactrotransferrin. In addition, we found an abundance of resistin.

Finally, in 2019, immunohistochemical analysis of the lacrimal sac MPC-Ds was performed and strong immunoreactivity for lysozyme was described. Few peripheral areas of concretions were positive for S100-A9, and all samples were negative for IgG [11]. Here, we show by the use of IF that IgG is, in fact, one of the most abundant antigens found. It was present in all samples studied, and predominantly localized on the surface, however, with high abundance in the core.

## 5. Conclusions

Taken together, here we examined the involvement of neutrophils and NETs in the formation and growth of human MPC-Ds. Considering the relatively low incidence of lacrimal sac obstruction with MPC-Ds, this study was constrained by a limited sample size. Despite the small size of our patient cohort, we ensured diversity by including participants from both genders and various age groups, thus providing a broader representation. We performed complete proteomic analysis by employing LC-ESI-MS and found neutrophil and NET markers in high quantities. We observed considerable NE activity in all tissues. By performing various stainings of all MPC-Ds, we determined the (co)localization of neutrophil markers (NE, citH3, and MPO), citrullination (citH3 and PAD4), immunoglobulins (IgG), and other types of leukocytes (hemoglobin for red blood cells and fibrinogen for thrombocytes). Through a multidisciplinary approach, we propose a classification system for mucopeptide concretions, with respect to the quantity of cellular and acellular material, polysaccharides, and mucosubstances. We describe the role of neutrophils as pivotal in the initial stages of dacryolithiasis, whereas NETs contribute to the later stages of disease pathogenesis, presumably maintaining the proinflammatory environment and impeding efficient clearance. These findings provide valuable insights into dacryolithiasis, highlighting the integral roles of both the innate and adaptive immune systems.

## Figures and Tables

**Figure 1 cells-12-01857-f001:**
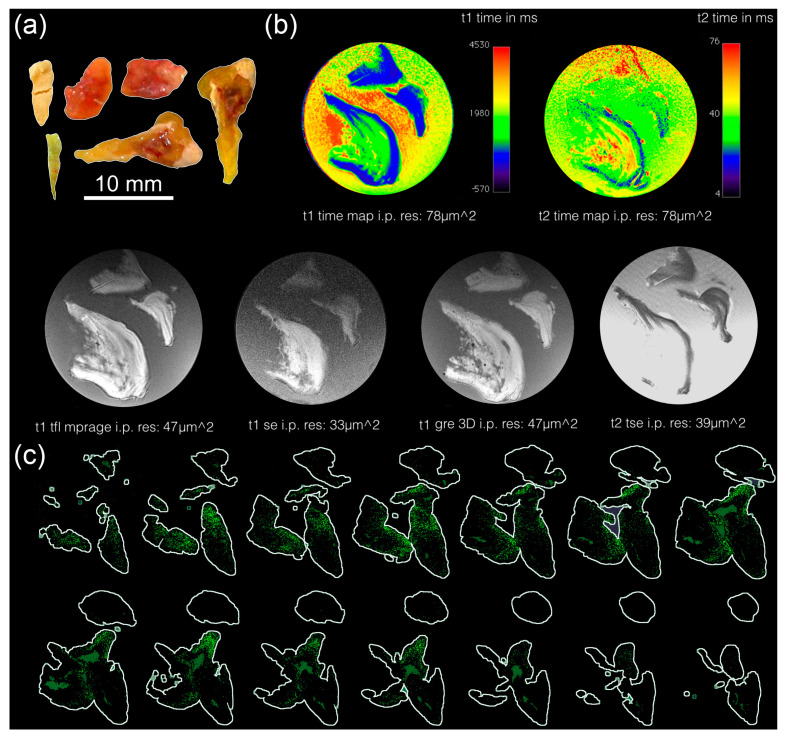
Macroscopic images, MRT, and µCT data of human MPC-Ds: (**a**) Macroscopic photograph of human MPC-Ds (*n* = 6); (**b**) Magnetic resonance tomography (MRT) and 3D surface reconstruction (grey) of a human MPC-D, t1: T−1 weighted, t2: T−2 weighted; (**c**) Microcomputed tomography (µCT) images of human MPC-Ds; electron dense inclusions are shown in green. Note human MPC-Ds are stiff, beige, yellow to orange in color, and appear like the inner cast of the lacrimal sac. MPC-Ds display like a layered structure and harbor only small calcified electron dense inclusions. In (**a**) and (**c**), the regions of interest are surrounded by automatically assigned white lines. Abbreviations: i.p. res: in-plane resolution.

**Figure 2 cells-12-01857-f002:**
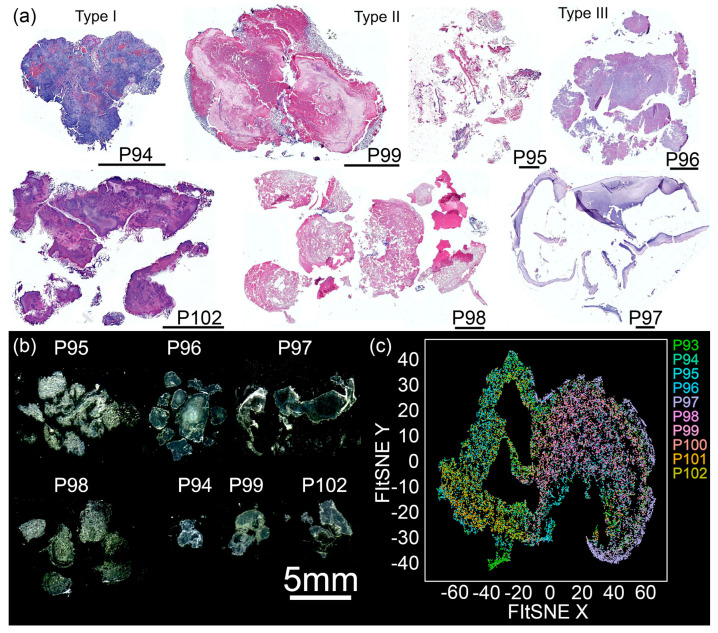
MPC-Ds are heterogenous but can be sorted into three major groups with respect to their morphology. (**a**) Hematoxylin and eosin (HE) staining of MPC-Ds (*n* = 7): P94, P99, P95, P96 (first row), P102, P98, and P97 (second row); vertically grouped based on staining: type I (left), type II (middle), and type III (right). Type I is characterized by a strong hematoxylin signal from chromatin or nuclei (blue), type II by a strong eosin signal from acellular material (pink), and type III consists of intermediate hematoxylin and eosin staining, from both chromatin/nuclei (blue) and non-nuclear material (pink). (**b**) Macrophotographs of paraffin-embedded MPC-Ds sections. Pictures were taken using the fluorescence scanner (Aperio Versa 8, Leica Biosystems). Samples were resized for easier comparison; all bars represent 2 mm. (**c**) FIt-SNE plot of PAS staining for all samples. Note the high pleomorphism of the MPC-Ds.

**Figure 3 cells-12-01857-f003:**
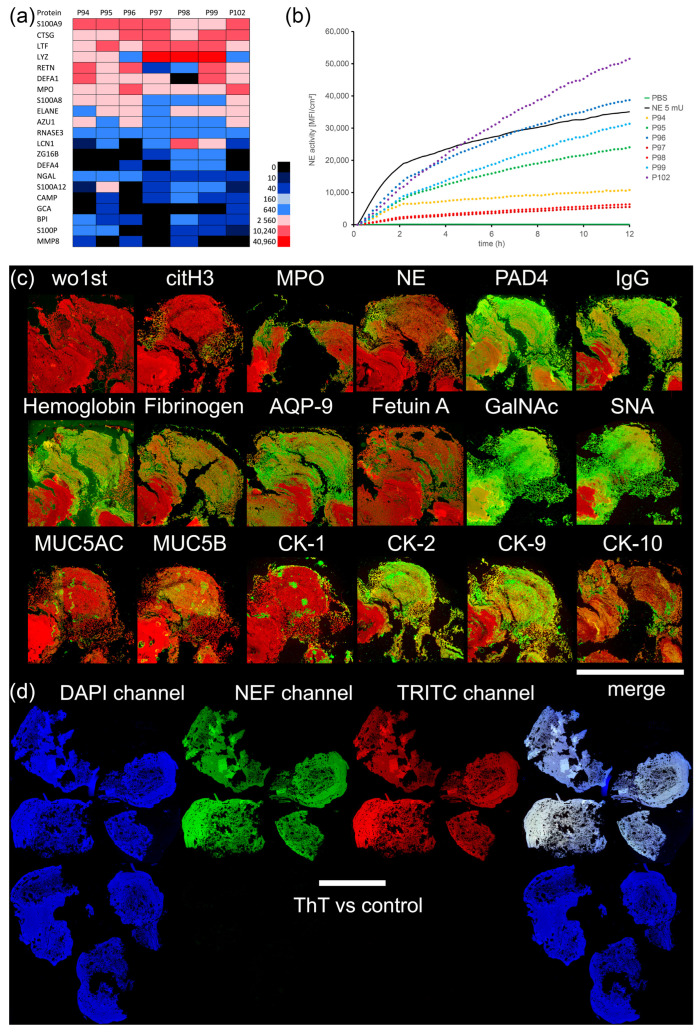
MPC-Ds are abundant in neutrophil-associated markers. (**a**) Proteomic analysis (LC-ESI-MS) of human MPC-Ds. Heat map displays MASCOT scores; (**b**) NE activity, normalized to sizes of tissues (MFI/cm2). NE (5 mU) served as positive control; (**c**) Representative immune fluorescence images of MPC-D (P99). All antigens are displayed in green, DAPI in red. The control was stained with DAPI and secondary antibody only (wo1st); (**d**) Representative image of mucopeptide concretion P98 stained with Thioflavin T (ThT). Upper row: ThT, lower row: control; bars represent 2 mm, and 4 mm, respectively. Note, neutrophil markers appear in all samples. NETs, characterized by the colocalization of citH3, MPO, or NE with DNA, preferentially appear on the surfaces. Abbreviations: QP-9: aquaporin 9, AZU1: azurocidin, BPI: bactericidal permeability-increasing protein, CAMP: cathelicidin antimicrobial peptide, CTSG: cathepsin G, citH3: citrullinated histone H3, CK: cytokeratin, DEFA1: neutrophil defensin 1, DEFA4: defensin alpha 4, ELANE: neutrophil elastase (NE), GalNAc: Galactose-β-1,3-GalNAc, GCA: grancalcin, IgG: immunoglobulin G, LCN1: lipocalin 1, LTF: lactoferrin, LYZ: lysozyme C, MFI: mean fluorescence intensity, MMP8: matrix-metalloproteinase 8, MUC: mucin, MPO: myeloperoxidase, NGAL: neutrophil gelatinase-associated lipocalin, PAD4: peptidyl arginine deiminase, RETN: resistin, RNASE3: eosinophil cationic protein, SNA: Galactose-β-2,6-Sialic acid, ZG16B: zymogen granule protein 16 homolog B.

**Figure 4 cells-12-01857-f004:**
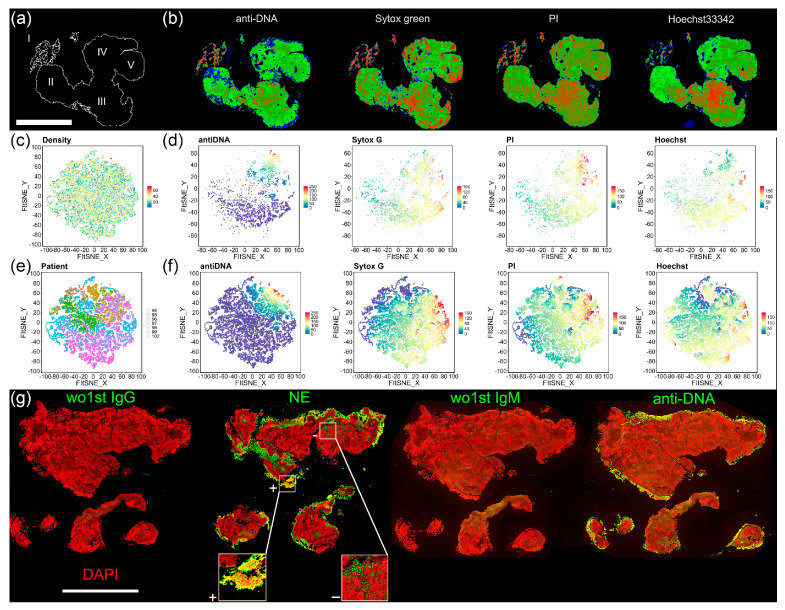
Neutrophils contribute to the formation and growth of MPC-Ds. (**a**) Overview of MPC-D P94, divided into 5 distinct regions (I–V); (**b**) Immune fluorescence images of MPC-D P94 stained with various DNA dyes (the signal was transformed into Rainbow RGB using Fiji); (**c**) FIt-SNE density plot indicating the automatic cluster assignments for all samples; (**d**) FIt-SNE plots of various DNA stainings for sample P99; (**e**) FIt-SNE plots as in (**c**), colored by patient ID and clustering DNA stainings from all patients (*n* = 7). Note, the high variability of individual samples; (**f**) FIt-SNE plots of each staining for all samples; bar represents 2 mm; (**g**) P102 stained for NE (green) and DNA (DAPI, red), or for extracellular DNA (anti-DNA, green). Images wo1st IgG, wo1st IgM served as controls for NE and anti-DNA, respectively. “+”: NET formation, “−”: no NET formation; bar represents 3 mm. Note NETs, preferentially stained by anti-NE and anti-DNA, can be found on the surfaces of MPC-Ds, and may possibly contribute to their inappropriate clearance.

**Table 1 cells-12-01857-t001:** Immune fluorescence of various antigens in MPC-Ds (average of all samples) ^1^.

	Core	Surface
MUC5B (57%)	0.14	0.29
Fetuin A (57%)	0.43	0.57
citH3 (43%)	0.14	0.71
CK-10 (100%)	0.71	1.00
NE (43%)	0.86	1.00
MPO (71%)	0.29	1.14
MUC5AC (86%)	0.57	1.14
CK-1 (86%)	0.57	2.00
Fibrinogen (86%)	0.86	1.86
AQP-9 (86%)	1.00	1.86
CK-9 (86%)	1.57	1.71
Hemoglobin (100%)	1.86	1.71
CK-2 (100%)	1.43	2.14
SNA (100%)	1.57	2.57
IgG (100%)	2.00	2.57
PAD4 (100%)	2.14	2.71
GalNAc (100%)	2.43	3.00

^1^ The intensity of various IF stainings was visually estimated for each MPC-D using a 0–3 point scale (no to high antigen abundance). Mean values from all samples (*n* = 7) were taken to construct the table, and the antigens were sorted according to their abundance. Parentheses display the percentage of samples in which certain antigens were detected.

## Data Availability

Data generated during the study are within the article or Appendix A.

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
