# Peer review of "Neutrophil Extracellular Traps Drive Dacryolithiasis"

_cells, 2023, doi:10.3390/cells12141857_

Round 1

Reviewer 1 Report

The manuscript "Neutrophil Extracellular Traps Drive Dacryolithiasis” by Leticija Zlatar and co-authors is a well written manuscript. The authors report the characterization of MPC-Ds from 7 clinical samples. The authors characterize these MPC-Ds using various methodologies and confirm 3 main types with respect to the presence of cellular and acellular material, the presence of polysaccharides, and mucosubstances. They observed heterogeneity in the presence of NET contents in these MPC-Ds but found a correlation between the presence of neutrophils and NETs.  The authors describe, and conclude that neutrophils play an earlier role and NETs might contribute later to the disease.

However, the paper suffers from some interpretation inconsistencies, and could be addressed in the text. The questions are below: 

Are the sections P94-P102 each represent a paraffin section from MPC-Ds? It is not clear whether the staining is due to different sections from paraffin embedded tissue. The correlation between the HE staining and NE activity analysis is confusing because the sections with more cellular staining P94 and P102 (type1) do not show similar NE activity patterns, and the sections P98 and P97 from type II and III show similar NE activity. If the authors claim that the staining pattern (blue) is conclusive of the presence of cells in P94, why is the NE activity low? The only results that correlate well are the sample P102 with more cell staining along with maximum NE activity. Although the authors discuss these discrepancies due to the tissue makeup, the thick network of amyloid fibers.

Is it also possible that the neutrophil markers such as MPO are leaky, and so NE activity is seen in some sections and not in others? In which case it may not be accurate to correlate the HE staining with NE activity.

It is possible if the authors analyzed more than 7 samples there will be some clarity in the pattern of heterogeneity such as age of person or the MPC-Ds formation.

Reviewer 2 Report

The authors of the present manuscript further investigated the role of neutrophil extracellular traps (NETs) in dacryolith formation and further characterized the composition of samples from 7 different patients. The manuscript is very well-written and reads well with a logical flow. The topic is of importance to the ophthalmic community as nasolacrimal duct obstruction is prevalent in our patient population and causes significant distress for many with many needing surgical intervention to correct. The methods are well-described and the study design is sound.

The authors could consider further discuss the etiology of the trauma/inciting events for dacryolith formation based on previous literature, such as the microbial role with actinomyces and staph/strep species that has been well-described.

Further discussing the significance of the different H/E staining patterns of the 3 different types of dacryoliths would also be helpful.

Otherwise I think that the manuscript is fit for publication as is.  

Reviewer 3 Report

Cells-2478842

The manuscript "The role of neutrophils and NETs in dacryolithogenesis" offers important new insights into the field of ophthalmology, particularly in the understanding of the formation of mucopeptide concretions, or dacryoliths. The multi-disciplinary approach used to understand the complex phenomenon of dacryolithogenesis is commendable. The use of micro-computed tomography, magnetic resonance tomography, histochemistry, mass spectrometry, and enzyme activity analyses indicate a comprehensive and detailed investigation.

The findings in this study which point towards the crucial role of neutrophils and Neutrophil Extracellular Traps (NETs) in the formation of these concretions could pave the way for new therapeutic strategies. The proposed model of gradual formation and growth of mucopeptide concretions is well thought out and reasonably justified, with neutrophils contributing to the initial stages of dacryolithogenesis and NETs to the later stages. The evidence for the involvement of the adaptive immune system later in the formation of these stones is also noteworthy and adds another dimension to the overall understanding of the pathogenesis.

In addition, the manuscript is clearly written in professional, unambiguous language. If there is a weakness, it is the samples size is relatively small and only seven mucopeptide concretions were removed by dacryocystorhinostom in this study. It would be worthwhile to follow up and enroll more patients to further study the underlying detailed mechanism of the macroscopic stones formation. 
